# Short-term impact of preschool sound exposure on outer hair cell function in young children: An analysis using pressurised distortion product otoacoustic emissions

Loisa Sandström[1]*, Sofie Fredriksson[1], Elina Mäki-Torkko[2], Mikael Ögren[3], Janina Fels[4], Kerstin Persson Waye[1]

**1** Sound Environment and Health, School of Public Health and Community Medicine, Institute of Medicine, University of Gothenburg, Gothenburg, Sweden, **2** Audiological Research Center, Faculty of Medicine and Health, Örebro University, Örebro, Sweden, **3** Department of Occupational and Environmental Medicine, Sahlgrenska University Hospital, Gothenburg, Sweden, **4** Institute for Hearing Technology and Acoustics, RWTH Aachen University, Aachen, Germany

\* loisa.sandstrom@gu.se

## Abstract

### Background

Preschool children are regularly exposed to high noise levels that may affect hearing. A previous study has linked preschool noise exposure to reduced distortion product otoacoustic emission (DPOAE) amplitudes. Since DPOAEs primarily reflect outer hair cell (OHC) activity, they provide an indirect marker of cochlear function. Measurement accuracy can be affected by middle-ear pressure. Pressurised DPOAEs (pDPOAEs) compensate for middle-ear pressure during recording.

### Methods

This cross-sectional study aimed to examine the relationship between preschool noise exposure and pDPOAE amplitudes while accounting for middle-ear pressure. Seventy-five children (4–6 years old) were monitored using dosimeters to measure the equivalent continuous sound level ($L_{AeqTi}$) and the 95th percentile of maximum levels ($L_{AFmax,95}$). Of these, 56 children completed pDPOAE testing at four time points during the preschool week. Linear mixed-effects models evaluated associations with noise exposure, time of day and progression across the week.

### Results

For personal dosimetry, the mean $L_{AeqTi}$ was 80 dB (range: 60–98 dB) and the mean $L_{AFmax,95}$ was 97 dB (range: 77–110 dB). Most $L_{AeqTi}$ levels (86.2%) were between 75–85 dB, with $L_{AFmax}$ levels exceeding 115 dB in 53.8% of the cases. No significant associations were found between $L_{AeqTi}$ or $L_{AFmax,95}$ and pDPOAE amplitudes

**Data availability statement:** The data underlying the results presented in this study cannot be shared publicly due to ethical restrictions related to research involving children, as specified in the approvals granted by the Swedish Ethical Review Authority (Dnr: 2019-03412, 2020- 03944, 2022-03580-02). A minimal anonymised dataset (excluding directly identifiable or sensitive information) has been deposited with the Swedish National Data Service (SND) and is available to qualified researchers who meet the criteria for access to confidential data. The dataset is accessible at: https://doi.org/10.5878/kp2z-2239. For further inquiries, or if to request to view the data send a requestan email to request@snd.se. researchdata@snd.se.

**Funding:** This study was funded by FORTE (Swedish Research Council for Health, Working Life, and Welfare), under the grant number 2018-00418. The research was conducted by Loisa Sandström, Kerstin Persson Waye, and their collaborators. The funding body had no role in the study design, data collection, analysis, decision to publish, or preparation of the manuscript.

**Competing interests:** The authors have declared that no competing interests exist.

(p > 0.05). Time-of-day differences were observed, with higher amplitudes in the afternoon at 4 kHz (p = 0.045) and 6 kHz (p = 0.047) in the right ear, and 3 kHz in the left ear (p = 0.021). Girls showed higher amplitudes than boys at 4 kHz in the left ear (p = 0.030).

## Conclusions

Although pDPOAE amplitudes varied with time of day, and sex, a direct exposure–response relationship with preschool noise was not demonstrated. Short-term variations in typical preschool noise exposure may not measurably affect cochlear function in young children. Future research should refine exposure assessment and recording protocols to reduce variability and improve detection of small physiological changes.

## Introduction

Sweden has one of the highest preschool participation rates in Europe, with 96.8% of four-year-olds and 97% of five-year-olds enrolled [1]. Preschools are universally accessible, offering a creative learning environments that support cognitive, social, and emotional development in alignment with the national curriculum [2]. While these settings support children's development and participation, their acoustic environment is often overlooked. Excessive noise in preschools can affect speech perception and learning [3] and may also have implications for children's auditory health.

Studies have linked preschool noise exposure to reductions in distortion product otoacoustic emission (DPOAE) amplitudes, suggesting potential alterations in outer hair cell function [3,4]. Preschools are characterised by elevated sound levels, primarily generated by children's speech, laughter, and play, with peak levels occurring during transitions, mealtimes, and group activities [5]. Unlike adults, children spend prolonged periods in these environments without access to quiet breaks and remain in close proximity to noise sources, increasing concerns about cumulative exposure.

Although occupational noise exposure among preschool personnel has been extensively documented, often exceeding regulatory limits [6], comparatively less attention has been given to the noise burden on children. As both primary sound sources and passive listeners, children may experience greater exposure levels than their teachers. Stationary noise measurements typically report $L_{Aeq}$ levels between 70 and 75 dB, with variations depending on room acoustics, group size, and activity type [7,8]. Personal dosimetry studies indicate that children's individual noise exposure frequently exceeds these estimates, with mean $L_{Aeq}$ levels reaching 85 dB before and 83 dB after an acoustic intervention, which is 6–8 dB higher than levels recorded for preschool personnel [9]. Recent findings further indicate that children's average $L_{AeqTi}$ during indoor preschool hours is 81 dB, with peak $L_{AFmax}$ levels reaching 112 dB [3]. These discrepancies highlight the need to differentiate ambient noise levels from individual exposure to ensure meaningful comparisons across studies employing different measurement methods.

Prolonged exposure to high sound environments may affect children's auditory health by contributing to temporary or permanent alterations in outer hair cell function, reducing otoacoustic emission amplitudes, and potentially increasing the risk of auditory fatigue or noise-induced hearing changes [3,4,10]. The World Health Organization (WHO) recommends limiting a yearly average from all leisure noise sources equivalent to ≤ 70 dB $L_{Aeq,24h}$ to prevent hearing damage [11]. Based on the equal energy principle, which states that the risk of hearing damage depends on the total accumulated sound energy over time [12] this recommendation corresponds to approximately 76 dB $L_{Aeq}$ for 6 hours per day over seven days a week. As equivalent noise levels in preschools often exceed this level, and as Swedish children typically spend full days in preschools, there may be reasons for concern. Given the continued maturation of central auditory processing and young children's limited ability to avoid or regulate environmental noise, potential long-term implications for auditory health in preschool settings warrant attention.

DPOAEs are sounds generated by the nonlinear interaction of two primary tones within the cochlea and are transmitted back through the middle ear to the ear canal, where they can be measured using a sensitive microphone. These emissions originate from the outer hair cells (OHCs) and provide a non-invasive, frequency-specific indicator of cochlear function [13–15]. OHCs enhance auditory sensitivity by amplifying low-level sounds and sharpening frequency selectivity via the cochlear amplifier [16].Loss of OHC function, as indicated by absent DPOAE responses, is typically associated with sensorineural hearing loss [17,18]. DPOAEs have been shown to decline following noise exposure, indicating short-term reductions in OHC-function and suggesting early signs of cochlear stress [4]. While such reductions, do not necessarily indicate permanent damage, they may present early-stage alterations in cochlear function [19].

DPOAE measurements depend not only on OHC function but also on middle ear status, as the middle ear serves as the transmission pathway for otoacoustic emissions. Negative middle ear pressure or otitis media can attenuate DPOAE amplitudes, particularly at low and mid frequencies, thereby complicating interpretations of cochlear responses [20]. This issue is particularly relevant for young children, who frequently experience transient middle ear conditions that may obscure DPOAE results. DPOAE testing provides an objective, non-behavioural assessment of peripheral auditory function. Unlike pure-tone audiometry, it does not require active participation, making it especially suitable for preschool-aged children [21].

To our knowledge only one previously published study has examined preschool noise exposure in relation to DPOAE amplitudes in children, but it did not systematically account for middle ear function, raising concerns about measurement accuracy [3]. Given the potential confounding effects of middle ear artefacts, it is essential to ensure that DPOAE recordings reflect actual function outer hair cell activity rather than conductive influences. To address this limitation, the current study used a pressurised DPOAE (pDPOAE) protocol, which involves adjusting ear canal pressure to match tympanic peak pressure. This approach reduces the impact of negative middle ear pressure, improving the reliability and validity of OAE measurements in paediatric populations.

## Study aims

This study aims to examine potential short-term effects in cochlear function in response to everyday sound exposure in preschool children by assessing DPOAE amplitude differences across different times of the day and over the preschool week in children aged 4–6 years. To improve methodological precision, pDPOAE testing and tympanometry assessment were incorporated at each measurement timepoint to account for middle ear status, which can influence DPOAE amplitudes.

Specifically, we assess whether time spent in preschool is associated with changes in cochlear function by comparing pDPOAE amplitudes between morning and afternoon sessions as well as between the beginning and end of the week. To quantify noise exposure, children were monitored using personal dosimeters measuring A-weighted equivalent continuous sound levels for time indoors ($L_{AeqTi}$) and fast-weighted 95 percentiles of maximum sound levels ($L_{AFmax,95}$). Additionally, we provide a descriptive comparison of individual noise exposure levels and stationary noise measurements to explore how well stationary recordings reflect children's personal noise exposure.

By targeting an under-researched age group and conducting measurements in real-world preschool settings, the study contributes to auditory risk research and supports ecological validity in assessing short-term effects on inner ear function.

## Materials and methods

### Study design and participants

These short-term repeated measures study was conducted between October 2022 and May 2024 in 15 preschool departments in Gothenburg, Sweden. It forms part of a longitudinal research project designed to investigate how the preschool sound environment influences children's cochlear function, perception of noise, and behavioral responses with plans to follow this cohort over time to examine potential long-term effects. A total of 100 children aged 4–6 years (47 boys and 53 girls) were enrolled. Of these, 75 participated in sound exposure monitoring using dosimeters, and 56 met the inclusion criteria for pDPOAE hearing function assessment.

### Ethical approval and consent

Ethical approval was obtained from the Swedish Ethical Review Authority (Reference No. 2019−03412, 2020−03944, 2022-03580-02). Written informed consent was obtained from each child's legal guardian or guardians. Participation was voluntary, and children could withdraw at any time without consequences.

### Eligibility and exclusion criteria

Participants were eligible if attending preschool for at least four days per week (08:30–15:00). Children were excluded if there was parent-reported hearing impairment, a history of ear disease or middle-ear surgery, middle-ear dysfunction as determined by tympanometry, or any acute illness on the test day (e.g., colds, flu, ear pain). Middle-ear dysfunction was excluded to prevent conductive hearing loss confounding the pDPOAE measurements, ensuring that observed amplitude variations reflected cochlear rather than middle-ear effects. A summary of participant inclusion is presented in Table 1.

### Individual sound level exposure

Individual sound exposure was assessed using Svantek SV104 dosimeters (Svantek, Warsaw, Poland) [22], which recorded sound pressure levels (SPL) by integrating A-weighted equivalent continuous sound levels (LAeq) over successive 10-second intervals throughout regular preschool activities. $L_{AeqTi}$ was computed to represent the continuous equivalent sound level over the full measurement period. $L_{AFmax,95}$ was selected instead of absolute $L_{AFmax}$ i.e., the maximum A-weighted sound pressure level using Fast time weighting) to mitigate the influence of isolated peaks (e.g., single events unrelated to the overall sound environment) and provide a more stable representation of high sound levels.

Dosimeters with ST 104C MEMS microphones (Type 2) complied with ISO 61672–1:2013 [23], which defines sound level meter performance for environmental noise assessment. Each device was placed on the left shoulder, a predefined position ensuring consistent exposure measurement near the ear while minimising movement artifacts and obstructions. This placement ensured consistency across participants and reduced variability due to movement or clothing interference.

Dosimeters recorded continuously during preschool hours on two measurement days per week: one at the beginning and one at the end of the preschool week. Calibration was performed before each use with a Class 1 Sound Calibrator (SV 36, Svantek, Warsaw, Poland; 94 dB SPL) [24] to ensure measurement accuracy. Data were processed using SvanPC++ software (version 3.4.19) [25].

### Hearing assessment

Outer hair cell function was assessed using pDPOAE, which involves measuring DPOAE amplitudes while temporarily adjusting ear canal pressure to match tympanic peak pressure. This procedure compensates for potential effects of

**Table 1. Participant Characteristics by Age and Sex enrolled cohort, N = 100).**

| Category | N | Boys | Girls |
|---|---|---|---|
| Ages | | | |
| 4 years | 44 | 22 | 22 |
| 5 years | 48 | 21 | 27 |
| 6 years | 8 | 4 | 4 |
| | | | |
| Age Summary | | | |
| Age Mean (years) | 5 | | |
| Age SD | 0.6 | | |
| Age Min | 4 | | |
| Age Max | 6.25 | | |
| | | | |
| **Measurements** | | | |
| Dosimeter (Noise Exposure) | 75 | | |
| pDPOAE (OHC – Function) | 56 | | |

*Note.* The table presents the total number of participants in the main study (N = 100) by sex and age. Age distribution is independent of sex. Dosimeter measurements (n = 75) assess individual noise exposure using individually worn devices. Pressurised Distortion Product Otoacoustic Emissions (pDPOAE) (n = 56) evaluate Outer Hair Cell (OHC) function based on predefined quality criteria.

the middle-ear pressure on the registration of DPOAE amplitudes as a measure of outer hair cell status. Testing was performed at four time points: morning and afternoon at both the beginning and end of the preschool week (Fig 1). This schedule was designed to evaluate intraday (morning vs. afternoon) and cumulative (beginning vs. end of week) differences in pDPOAE amplitudes, following an established protocol [3].

Assessment was conducted twice weekly, once at the beginning and once at the end of the week. On both days, children wore individual dosimeters (green boxes) to monitor personal sound exposure levels. Cochlear function was assessed using pressurised distortion product otoacoustic emissions (pDPOAE; dark blue boxes), following tympanometry (light blue boxes) to evaluate middle ear status. Dashed arrows indicate that hearing assessments were scheduled on the same days as dosimeter use, although participation varied. Solid arrows denote the sequence of hearing assessments, with tympanometry conducted prior to pDPOAE measurements.

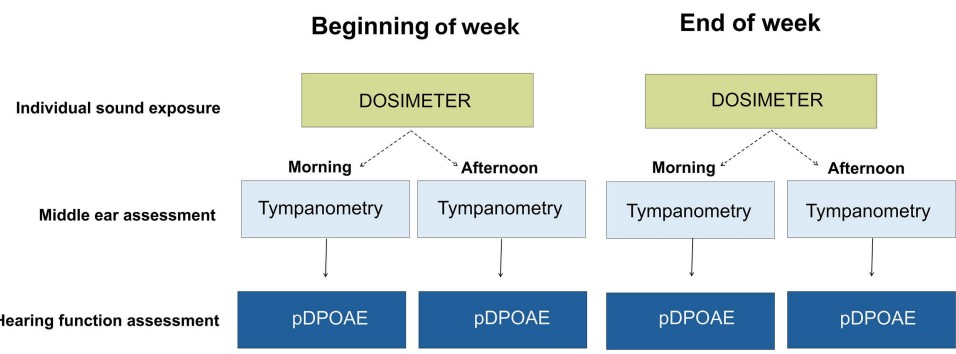

**Fig 1. Measurement protocol for sound exposure and hearing function assessment.**

pDPOAE testing was conducted using the Interacoustics Titan 440 system (Interacoustics, Middelfart, Denmark) [26] with Titan Suite software version 3.6.1. [27]. Two primary stimulus tones were used: L1 = 65 dB SPL and L2 = 55 dB SPL (where dB SPL refers to sound pressure level in the ear canal), with an f2/f1 ratio of 1.22. The amplitude of the distortion product was analysed in dB SPL at the frequency $2f_1 - f_2$, consistent with standard DPOAE protocols. pDPOAE amplitudes were recorded at six frequencies (3, 4, 5, 6, 7, and 8 kHz) to assess early signs of noise-induced hearing loss (NIHL), which typically affects frequencies ≥ 3 kHz, with 4 kHz being the most vulnerable in adults [28]. A susceptibility to higher frequencies (5–8 kHz) in children was hypothesised based on anatomical differences in ear canal resonance and head-related transfer functions (HRTFs) [29]. Pressure compensation was achieved by adjusting the probe pressure to match the tympanic peak pressure, based on tympanometric results, prior to each pDPOAE measurement [20]. Whenever possible, a designated testing station was set up in a small office or a staff-designated room, ensuring that measurements were conducted under consistent conditions with minimal ambient noise.

To ensure valid pDPOAE measurements, otoscopy was performed prior to testing using a Heine Mini 3000 otoscope (HEINE Optotechnik, Herrsching, Germany) [30] to check for cerumen or obstructions. Tympanometry was conducted using the same equipment as for pDPOAE testing, using a 226 Hz probe tone at 65 dB SPL across a pressure range of −300 to +200 daPa at a rate of 50 daPa/s [31].

## Stationary sound level measurements

To assess overall sound levels in the preschool environment, stationary measurements were conducted using ceiling-mounted Svantek SV104 dosimeters (Svantek, Warsaw, Poland) [22]. These dosimeters recorded A-weighted SPL as 10-second time-averaged values, which were subsequently used to compute $L_{AeqTi}$ and $L_{AFmax}$ over the full measurement period.

Dosimeters were suspended at 180 ± 20 cm from the floor — a consistent height selected to minimise floor reflections and avoid interference with preschool activities. Devices were mounted in open woven baskets that did not obstruct or distort sound. To reduce reflections and external noise, dosimeters were placed at least 1 m from walls and kept away from corners, windows, doors, and ventilation systems. Placement was adjusted as needed to reflect typical child activity areas and ensure measurements were representative of actual noise exposure. A uniform placement protocol was followed across all preschools to ensure comparability despite differences in room layout and ceiling height.

Sound levels were recorded continuously during preschool hours (07:00–17:00) across five consecutive weekdays (Monday to Friday). Measurements were taken in two rooms per preschool whenever possible: (1) a larger activity room, where most activities and mealtimes occurred, and (2) a smaller room or atelier used for additional activities. Analyses were restricted to 08:30–14:30 to align with scheduled indoor activities.

Dosimeters were calibrated before each measurement period and recalibrated midweek when devices were exchanged due to battery and memory limitations.

## Data inclusion criteria

Predefined inclusion and exclusion criteria were applied for both individual sound exposure and pDPOAE measurements. For individual sound exposure, dosimeter recordings were included only for regular indoor preschool activities, excluding data collected during lunchtime and outdoor periods. Data were collected during periods when children were active. Nap time was not scheduled in the participating preschool departments, although the early afternoon typically involved calmer periods. This approach ensured that the analysis focused on typical indoor sound exposure. After applying these criteria, 130 valid measurements were retained for both $L_{AFmax,95}$ and $L_{AeqTi}$.

To assess outer hair cell function, a stepwise exclusion process was applied. Firstly, otoscopy was used to exclude ears with visible cerumen. Second, tympanometry was conducted to evaluate middle ear function. If results indicated middle ear dysfunction, defined as not meeting the criteria for type A tympanogram (tympanic peak pressure between −150 daPa

to +50, or static admittance was below 0.3 mmho), the corresponding pDPOAE measurement for that ear and timepoint were excluded. This step was taken to reduce potential confounding related to middle ear status.,.

For ears that passed tympanometry, pDPOAE measurements were further evaluated for signal quality at the frequency level. Recordings were retained only if they met all predefined thresholds: a signal-to-noise ratio (SNR) of at least 6 dB, an amplitude of at least –10 dB SPL, and a measurement reliability of at least 80%, as indicated by the recording software. These criteria were applied independently for each of the six test frequencies in both ears and are consistent with the procedure described by Sandström et al. [3].

For stationary sound level measurements, data were analysed within the 07:30–14:30 timeframe. This cutoff was selected based on observed noise patterns, as sound levels before and after this period were typically below 45 dB, indicating the absence of children and staff. This selection aligns with preschool routines, where most departments begin daily activities around 08:00 with breakfast and transition outdoors in the afternoon, ensuring that measurements reflect typical indoor sound exposure patterns.

## Statistical analysis

All statistical analyses were conducted in R (version 4.x; R Core Team, 2024) using lme4 (version 1.1−36) [32] for model estimation, lmerTest (version 3.1−3) [33] for statistical inference, and ggplot2 (version 3.5.1) [34] for visualisation.

## Descriptive statistics

Descriptive analyses were performed to summarise individual sound exposure ($L_{AeqTi}$, $L_{AFmax,95}$), pDPOAE amplitudes, and stationary sound levels. Summary statistics including mean, standard deviation, range, and percentiles were computed for all primary outcome variables. Additional descriptive summaries were generated for pDPOAE amplitudes across ears and frequencies, and for stationary environmental measurements in the preschool setting.

## Linear mixed-effects models

Linear mixed-effects models (LMMs) were used to assess the effects of noise exposure, time of day, and week progression on pDPOAE amplitudes while accounting for repeated measures within individuals. Two separate sets of models were used for analysis: the Sound Exposure Models, which examined associations between pDPOAE amplitudes and noise exposure ($L_{AeqTi}$ and $L_{AFmax,95}$) in separate models, and the Time Effects Model, which assessed how pDPOAE amplitudes varied across time of day and week progression, including an interaction between these two factors. All models included participant ID as a random intercept to account for repeated measures and within-subject variability. An overview of these models is provided in Table 2.

**Table 2. Summary of statistical models for pDPOAE amplitude analysis.**

| Model type | Purpose | Fixed effects | Random effects | Outcome variable |
|---|---|---|---|---|
| **Sound Exposure Models** ($L_{AeqTi}$ & $L_{AFmax,95}$) | Sound exposure and pDPOAE amplitude associations | $L_{AFmax,95}$, $L_{AeqTi}$, day of the week, sex | Participant-level random intercepts | pDPOAE amplitudes |
| **Time Effects Model** | Assess pDPOAE amplitude variation by time of day and week. | Time of day, day of the week, time × day interaction, sex | Participant-level random intercepts | pDPOAE amplitudes |
| **Model Diagnostics** | Validate assumptions | Residuals vs. fitted values, QQ plots | N/A | Model fit criteria (AIC, BIC, ICCs) |

*Note.* The Sound Exposure Model consists of two separate models, one examining the association between pDPOAE amplitudes and $L_{AeqTi}$ and another examining pDPOAE amplitudes and $L_{AFmax_{95}}$. Both models adjust for sex and day of the week and include participant-level random intercepts. The Time Effects Model evaluates variations in pDPOAE amplitudes across time of day and week progression, including a time × day interaction term, while also adjusting for sex. Model diagnostics assess normality and homoscedasticity, with model fit evaluated using AIC, BIC, and ICCs.

In the Sound Exposure Models, separate models were run for $L_{AeqTi}$ and $L_{AFmax,95}$ as independent predictors. These models adjusted for sex as a potential confounder and linked each pDPOAE value to the corresponding weekday. Only afternoon measurements were included, as they reflect actual daily exposure, while morning data were excluded because they represent baseline conditions prior to significant sound exposure.

The Time Effects Model evaluated the influence of time of day (morning vs. afternoon) and week progression (beginning vs. end of the week) on pDPOAE amplitudes. This model also included a time × day interaction effect and adjusted for sex as a potential confounder.

Model selection was based on the Akaike Information Criterion (AIC) and Bayesian Information Criterion (BIC), with ΔAIC used to compare model fit. Intraclass correlation coefficients (ICCs) were computed from the base model to assess the proportion of total variance attributable to between-participant variability in pDPOAE amplitudes. ICC values were interpreted as follows: poor reliability (<0.50), moderate reliability (0.50–0.75), and high reliability (>0.75) [35].

### Descriptive within-subject changes in pDPOAE amplitudes

We summarised within-subject pDPOAE amplitude differences across two intervals (morning versus afternoon, beginning versus end of week) as descriptive context only. Session-level means were computed per participant, ear and frequency, and change scores were derived between the relevant timepoints. These summaries were not used for inference and are presented in the Supplement.

### Supplementary analyses

Additional descriptive analyses addressed tympanometric peak pressure (TPP) values among retained recordings. Histograms and scatterplots (S3–S5 Figs) display the distribution and within-subject variability of TPP for both ears and all measurement rounds. Table C in S1 File Supporting Table summarises the percentage and count of retained recordings by TPP category (≥–50 daPa, –49 to –1 daPa, 0 to +50 daPa).

### Model diagnostics

Model assumptions were evaluated using residual vs. fitted plots and quantile-quantile (Q-Q) plots to assess homoscedasticity and normality. Diagnostic results (Figs A–L in S2 File Diagnostic Plots) confirmed that model assumptions were met.

## Results

### Descriptive data

Children's individual sound exposure was assessed using personal dosimetry, measuring $L_{AeqTi}$ and $L_{AFmax,95}$. Summary statistics, including the number of valid recordings, ranges, means, and confidence intervals, are presented in Table 3.

The mean $L_{AeqTi}$ was 80.4 dB (range: 60.1–98.4 dB), with 86.2% of measurements between 75 and <85 dB(A), and 60% above 80dB range. $L_{AFmax,95}$ levels averaged 97.1 dB (range: 76.9–109.7 dB), with 94.6% below 105 dB. Among the remaining 5.4%, most values (4.6%) ranged from 105–109 dB, while 0.8% reached ≥110 dB. In contrast, absolute $L_{AFmax}$ values were substantially higher, with 53.8% exceeding 115 dB, indicating frequent peak noise events. Recording durations averaged 5 hours and 7 minutes (range: 42 minutes to 7 hours and 18 minutes).

Table 4 provides a detailed categorization of measurements for $L_{AeqTi}$, $L_{AFmax,95}$, and $L_{AFmax}$, showing the number and percentage of valid recordings within defined exposure intervals for each metric. Fig 2 presents the distribution of $L_{AeqTi}$ and $L_{AFmax,95}$ across participants. Table 5 compares individual-level dosimetry data with stationary sound level measurements from participating preschools.

Scatter plot showing $L_{AeqTi}$ (equivalent continuous sound level; blue circles) and $L_{AFmax,95}$ (95th percentile of maximum sound levels; orange triangles) recorded from individually worn dosimeters. The y-axis displays sound pressure level

**Table 3. Summary statistics for individual sound exposure measurements in children using individually worn dosimeters.**

| | N | Min | Max | Mean | Std. Dev. | 95% CI |
|---|---|---|---|---|---|---|
| dB $L_{AeqTi}$ | 130 | 60 | 98 | 80 | 6 | 79-81 |
| dB $L_{AFmax_{as}}$ | 130 | 77 | 110 | 97 | 4.9 | 96-98 |
| Log Time | 130 | 0:42 | 7:18 | 5:07 | 1:20 | 4:53-5:21 |

*Note.* This table presents key statistics for $L_{AeqTi}$ (continuous equivalent noise exposure) and $L_{AFmax95}$ (95th percentile maximum noise level, Fast), including the number of valid measurements (N), minimum and maximum recorded values (Min, Max), mean exposure levels (Mean), standard deviation (Std. Dev.), and 95% confidence intervals (95% CI). Log time represents the duration of noise exposure measurements recorded indoors and is presented in hh:mm format.

**Table 4. Distribution of measurements (n) percentage of total) within categories of exposure intervals in $L_{AeqTi}$, $L_{AFmax,95}$ and $L_{AFmax}$.**

| | $L_{AeqTi}$ | | $L_{AFmax,95}$ | | $L_{AFmax}$ | |
|---|---|---|---|---|---|---|
| < 75 dB | (18) | 13.9% | | | | |
| 75-79 dB | (34) | 26.2% | | | | |
| 80-84 dB | (54) | 41.5% | | | | |
| ≥85 dB | (24) | 18.5% | | | | |
| <105 dB | | | (123) | 94.6% | (6) | 5.8% |
| 105-109 dB | | | (6) | 4.6% | (16) | 15.4% |
| 110-114 dB | | | (1) | 0.8% | (26) | 25% |
| ≥ 115 dB | | | | | (56) | 53.8% |

*Note.* This table shows the distribution of $L_{AeqTi}$, $L_{AFmax,95}$, and $L_{AFmax}$ across noise exposure levels. $L_{AeqTi}$ is the equivalent continuous A-weighted sound level over a time interval. $L_{AFmax95}$ is the 95th percentile of $L_{AFmax}$, indicating the level exceeded 5% of the time, while $L_{AFmax}$ represents the peak A-weighted sound level. The table includes the number (n) and percentage of valid measurements in each noise level category.

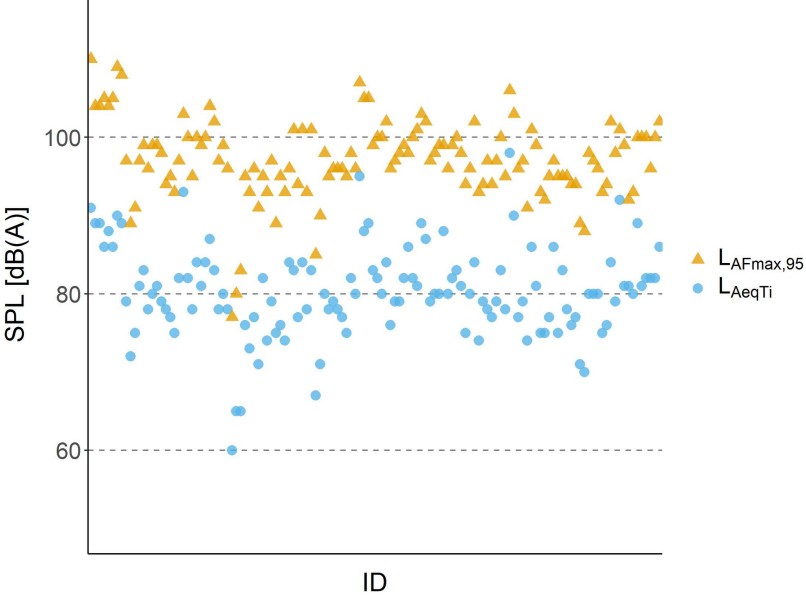

**Fig 2. Individual sound exposure levels measured with dosimeters.**

Table 5. Comparison of individual and stationary dosimeter measurements in preschools.

| Percentiles (%) | Individual measurements | | | | | | Stationary measurements | | | | | |
| --- | --- | --- | --- | --- | --- | --- | --- | --- | --- | --- | --- | --- |
| | $L_{AeqTi}$ (dB) | | | $L_{AFmax}$ (dB) | | | $L_{Aeq}$ (dB) | | | $L_{AFmax,95}$ (dB) | | |
| | Mean | Range | | Mean | Range | | Mean | Range | | Mean | Range | |
| 5 | 70 | 60 | 74 | 102 | 96 | 106 | 60 | 58 | 63 | 94 | 91 | 97 |
| 50 | 80 | 80 | 81 | 115 | 114 | 116 | 66 | 66 | 66 | 103 | 103 | 103 |
| 95 | 92 | 89 | 98 | 123 | 123 | 124 | 71 | 71 | 71 | 118 | 110 | 124 |

*Note.* Individual measurements reflect data from dosimeters worn by children, while stationary measurements were recorded using fixed-location dosimeters placed in preschool environments. $L_{AeqTi}$ refers to the equivalent continuous A-weighted sound level for Time indoors and $L_{AFmax}$ represents the maximum A-weighted sound level measured with a fast time weighting. Values are presented for the 5th, 50th, and 95th percentiles, with corresponding means and observed ranges (minimum–maximum) for each percentile.

(SPL) in dB(A); the x-axis corresponds to individual participants. Dashed horizontal lines mark reference levels at 60, 80, and 100 dB(A). The spread of points illustrates inter-individual variation in both continuous and peak sound exposures.

## pDPOAE dataset and retention

Retention rates, based on the number of frequency-specific pDPOAE measurements, were higher at the beginning of the week (n = 457 [56.4%] in the Monday morning session and n = 446 [57.3%] in the afternoon) and declined toward the end (n = 375 [47.1%] on Thursday morning and n = 350 [45.6%] in the afternoon) (Table B in S1 File Supporting Tables).

Tympanometric peak pressure (TPP) was measured prior to each pDPOAE session, and only recordings within the inclusion range (−150 to +50 daPa) were retained. Fig 1 shows a broad distribution of TPP values in both ears, with most observations clustering near −50 daPa and a secondary peak around 0 daPa. Within-individual changes in TPP between baseline (Round 1) and follow-up rounds (Rounds 2–4) are shown in S4 Fig; while many values remained stable across sessions, considerable variation between sessions was also observed.

## pDPOAE measurements

pDPOAE amplitudes ranged from −9.5 to 19.9 dB SPL, with the greatest variability observed at 7–8 kHz, as reflected in larger standard deviations (Table D in S1 File Supporting Tables). Group means were consistently higher in the right ear across all test frequencies. Mean amplitude values by frequency and measurement condition are summarised in Table D in S1 File Supporting Tables. Fig 3 visualises pDPOAE amplitudes by ear and frequency, comparing measurements from the beginning and end of the preschool week.

(dB SPL) for the right and left ears across frequencies from 3 to 8 kHz, measured at the beginning and end of the preschool week. Each panel displays data for one ear, with red lines representing morning measurements and blue lines representing afternoon measurements. Mean amplitudes peaked at 5–6 kHz and declined at higher frequencies in both ears. No error bars are shown.

## Effects of sound exposure on pDPOAE amplitudes

Weak but statistically significant correlations were observed between pDPOAE amplitude and both $L_{AeqTi}$ and $L_{AFmax,95}$ in the right ear ($L_{AeqTi}$: r = 0.27, p = 0.023; $L_{AFmax,95}$: r = 0.26, p = 0.028), but not in the left ear (Fig 4). These associations were not confirmed in the linear mixed-effects models, and no consistent exposure–response pattern emerged across ears or frequencies.

## Scatter plots display the relationship between pDPOAE amplitudes

(x-axis; dB SPL) and individual sound exposure levels (y-axis; dB(A)) or afternoon measurements at both the beginning and end of the preschool week. Panels are ordered right ear first, then left ear. Exposure is represented by $L_{AeqTi}$ (blue

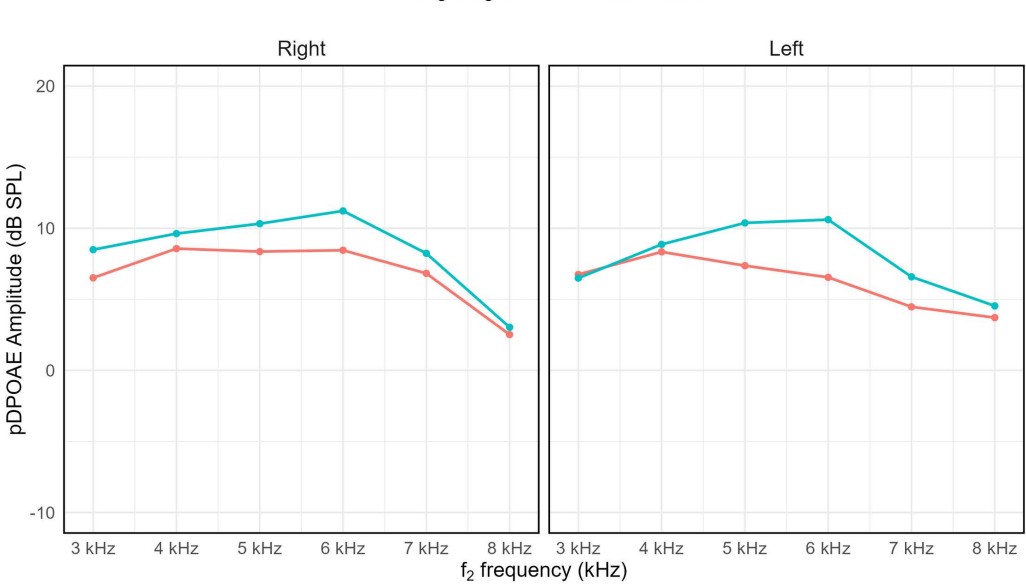

**Fig 3. Mean pDPOAE amplitudes.**

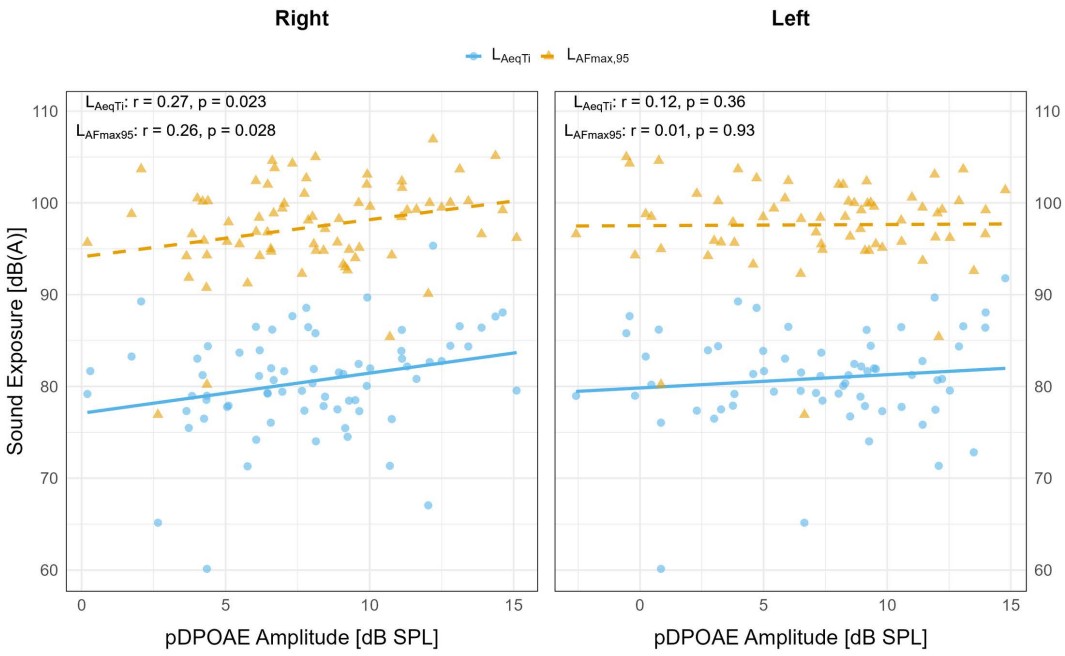

**Fig 4. Associations between individual sound exposure and pDPOAE amplitudes in left and right ears.**

circles) and $L_{AFmax,95}$ (orange triangles). Solid lines indicate linear regression trends for $L_{AeqTi}$; dashed lines represent trends for $L_{AFmax,95}$. Pearson correlation coefficients and p-values shown are based on unadjusted bivariate associations and are presented for descriptive purposes only. Inferential statistics were obtained from linear mixed models.

A significant sex difference was observed at 4 kHz in the left ear, with girls exhibiting higher amplitudes than boys. This effect remained significant in separate models controlling for $L_{AFmax,95}$ (β = 2.807, 95% CI [0.39, 5.22], p = 0.030) and $L_{AeqTi}$ (β = 2.782, 95% CI [0.40, 5.17], p = 0.030). Full model outputs are presented in Table E and F in S1 File Supporting Tables.

### Time-of-day and end-of-week variability in pDPOAE amplitudes

pDPOAE amplitudes varied significantly with both time of day and week progression. Afternoon measurements were significantly higher than morning measurements at 4 kHz (β = 1.327, 95% CI [0.03, 2.62], p = 0.045) and 6 kHz (β = 1.380, 95% CI [0.02, 2.74], p = 0.047) in the right ear, as well as at 3 kHz in the left ear (β = 1.152, 95% CI [0.18, 2.12], p = 0.021).

Additionally, amplitudes increased significantly from the beginning to the end of the week at 5 kHz in the right ear (β = 2.390, 95% CI [0.20, 4.58], p = 0.033). A significant interaction between within day and week progression was observed at 8 kHz in the left ear (β = 4.470, 95% CI [1.23, 7.71], p = 0.008).

Sex had a significant effect on pDPOAE amplitudes at 5 kHz in the right ear, where girls exhibited higher amplitudes than boys (β = 2.915, 95% CI [0.17, 5.66], p = 0.037). Full model outputs are presented in Table G in S1 File Supporting Tables.

A descriptive distribution of raw, within-subject changes in pDPOAE amplitude is provided in S5 Fig.

## Discussion

This study investigated how real-world preschool noise exposure relates to short term effect on cochlear function in young children, using pDPOAE as a non-invasive, objective indicator of outer hair cell status. Although significant variation in pDPOAE amplitudes was observed across time of day and over the preschool week, no direct associations emerged between individual noise exposure levels ($L_{AeqTi}$, $L_{AFmax,95}$) and cochlear responses. This suggests that while temporal factors influence pDPOAEs, their effects may not be fully accounted for by short-term fluctuations in environmental noise, as measured by personal dosimetry.

Afternoon and end-of-week measurements were associated with significantly higher amplitudes at specific frequencies, particularly in the right ear at 4 and 6 kHz, and in the left ear at 3 kHz. Higher DPOAE amplitudes are generally considered indicative of better outer hair cell function. The present findings contrast with previous research showing amplitude reductions following noise exposure [4,10]. Such reductions are typically interpreted consistent with temporary changes in cochlear OHC-function. This pattern has also been observed in a study involving preschool-aged children. Sandström et al. [3] reported DPOAE amplitude reductions at 6 and 8 kHz in the right ear in relation to $L_{AeqTi}$. In addition, reductions at 3 and 4 kHz in relation to bot $L_{AeqTi}$, and time spent at preschool. A likely explanation for this discrepancy is variation in measured noise exposure levels and other methodological differences between studies, including differences in how exposure was characterised (for example, use of $L_{AFmax,95}$ instead of $L_{AFmax}$, and longer measurement durations), as well as variations in DPOAE measurement protocols.

The present study, based on 130 individual noise exposure measurements, recorded a mean $L_{AeqTi}$ of 80.4 dB, with 18.5% of values exceeding 85 dB. In contrast, Sandström et al. [3] reported a slightly higher mean $L_{AeqTi}$ of 81 dB and a considerably greater proportion (30%) of measurements exceeding 85 dB. While both studies documented $L_{AFmax}$ values over 100 dB, notable differences were also observed in the highest $L_{AFmax}$ category with 53.8% of measurements exceeded 115 dB, compared to 45% in Sandström et al. [3]. These findings indicate that maximum noise levels in the current study were not lower than expected but rather higher, highlighting the presence of intermittent, high-level noise events in preschool environments. No significant reduction in pDPOAE amplitude was observed in relation to these maximum exposures at the group level, which may in part reflect statistical limited power.

Despite these level variations between this study and [3], no significant associations were found between individual $L_{AeqTi}$ or $L_{AFmax,95}$ and pDPOAE amplitudes. The absence of significant exposure-response associations may reflect the subtlety or transient nature of any noise-induced cochlear changes at the observed exposure levels. In adult populations,

amplitude reductions often recover within 24 hours [4,10] and younger children may show either increased resilience or different recovery dynamics. The statistical power of the present study may have been insufficient to detect subtle or rapidly reversible changes, particularly given the exclusion of a substantial number of measurements. Furthermore, pDPOAEs may not be sufficiently sensitive to detect small or rapidly reversible changes in outer hair cell function in naturalistic preschool settings. It is also plausible that relevant exposure effects occurred earlier in the day or week, prior to the specific pDPOAE measurement, and were not captured in the corresponding dosimetry–pDPOAE association.

The one-week data collection period was designed to assess whether short-term variability in preschool noise exposure, both within days and across the week, corresponds to measurable variation in cochlear responses. Longer-term, cumulative effects would require extended follow-up, which was beyond the scope of this short-term observational study. A further limitation is that, while general listening habits outside of preschool were reported by caregivers at baseline, the absence of detailed, time-specific data on out-of-school noise or music exposure limits our ability to fully account for all sources of potential variance in cochlear outcomes during the study period. These background data were collected only to provide descriptive context, were not time-linked to the period of noise exposure assessment and were not included in the primary analyses. For this reason, they were not presented in the main results. The detection (or absence) of within-week variation may still have implications for understanding how repeated exposure might influence auditory development over time. Future studies should include precise, time-specific measures of out-of-preschool noise and music exposure to allow for more comprehensive assessment of total auditory load.

A key methodological strength of this study was the use of pDPOAEs, selected based on prior evidence and theoretical rationale suggesting improved validity under fluctuating middle-ear conditions [36,37]. Matching the probe pressure to the tympanometric peak pressure helps restore DPOAE amplitudes that would otherwise be attenuated by negative middle-ear pressure, particularly at lower frequencies [20]. For sessions where TPP was close to 0 daPa, pDPOAE and standard DPOAE would be expected to yield similar results. Given that a substantial proportion of included recordings exhibited mild-to-moderate negative pressure, the use of pDPOAE provided a clear advantage for this sample. Although pressurisation primarily benefits responses below 2000 Hz, Sun and Shaver [38] also reported measurable improvements at 3 kHz, indicating that middle-ear status can still influence responses in this range. No adverse effects on higher frequencies have been reported [36], supporting our decision to apply it across the 3–8 kHz range. This approach is particularly relevant in field studies involving children, where transient middle-ear dysfunction is common and can otherwise confound the interpretation of outer hair cell function.

pDPOAE measurements, in which probe pressure is matched to the tympanometric peak pressure, mitigate attenuation from negative middle-ear pressure and support valid inference on outer hair cell status under fluctuating middle-ear conditions [36,37]. Immediate test–retest reliability was not assessed in this study. This decision was made to minimise procedural burden and avoid unnecessary repetition for participating children, who were already required to complete up to four pDPOAE assessments in a single week. Conducting additional repeated tests was not considered ethically or practically appropriate in this age group. To reduce technical variability, we implemented consistent tympanometric screening, standardised probe placement, and pressurised measurements across all sessions.

In this study, ears with tympanometric peak pressure outside the defined range were excluded prior to assessment of pDPOAE signal quality. This exclusion step had the greatest impact on data retention, as seen in Table A in S1 File Supporting Tables.. Approximately one third of measurements were excluded at this step. For ears that passed tympanometry, additional frequency-level exclusions were applied based on predefined signal quality criteria. These criteria: signal-to-noise ratio, amplitude and reliability, contributed to further reductions, but to a lesser extent. For example, in the right ear at 5–6 kHz, 78–80 measurements were excluded due to tympanometry, compared to 14–23 excluded across all subsequent steps. A similar pattern was observed in the left ear, with 88–90 excluded at tympanometry and 13–25 excluded thereafter. As shown in Table B in S1 File Supporting Tables, the percentage of retained responses after applying exclusion criteria, including tympanometry and signal quality, declined over the week (from ~56% to ~46%) and was

lower in the afternoon. This pattern may reflect increased variation in middle-ear status, as well as reduced compliance later in the day, such as restlessness or difficulties maintaining probe placement. Thompson et al. [37] reported more negative middle-ear pressure in adults during afternoons, suggesting similar diurnal variation could occur in children, leading to higher exclusion rates. Additionally, mild subclinical middle-ear changes and decreased child engagement or movement artefacts in later sessions may have contributed to unstable probe placement and higher exclusion rates despite similar measurement attempts [20,36]. Together, these factors likely reduced the proportion of analysable measurements toward the week's end.

Because entire sessions were excluded based on tympanometric results, the retained dataset may over-represent children with more stable middle-ear function, particularly toward the end of the week. This could have led to a subtle upward shift in average pDPOAE amplitude, not due to physiological change, but because measurements from children with poorer emission quality or borderline middle-ear function were systematically excluded. This selection effect may have biased the amplitude estimates upwards. This interpretation is supported by the observed decline in overall data retention later in the week (Table B in S1 File Supporting Tables) and by the broad distribution of tympanometric peak pressures among retained sessions (S3 Fig). Most exclusions were due to tympanometry, not pDPOAE-specific criteria. While many children showed stable TPP values across sessions, substantial within-individual variation was also observed (S4 Fig).

This trade-off reflects a methodological decision to prioritise valid emissions over completeness. Prior studies on young adults have shown that short-term noise exposure can reduce DPOAE amplitudes within the measurable range but rarely eliminates emissions entirely [10,39]. These procedures were therefore necessary to limit artefacts while preserving the ability to detect subtle within-child variation.

Sex-related differences in pDPOAEs were also explored, as prior studies have consistently reported higher amplitudes in girls [40–42]. In the present study, girls showed slightly higher amplitudes at 4 kHz in the left ear, and a sex-by-time interaction emerged at 4 kHz in the right ear. These effects were isolated and inconsistent across models, raising the possibility of random variation, individual noise exposure differences, or measurement artefacts (e.g., probe fit, middle-ear status). While our results are broadly consistent with the literature, the inconsistencies suggest that sex-related effects in pDPOAEs may be subtle and influenced by measurement conditions, particularly in young children. Further research with larger samples and appropriate statistical corrections is needed to clarify how sex and time-related factors may interact to influence cochlear function in early childhood.

Another potential explanation for the lack of exposure-response associations lies in the use of A-weighted sound level estimates. A-weighting aligns with adult auditory sensitivity by attenuating low and high frequencies, potentially underestimating sound pressure levels at higher frequencies that may be relevant to young children. Despite these limitations, A-weighting was used in this study because it remains the international standard for environmental noise measurement and facilitates comparison with previous research. Anatomical differences, such as shorter ear canals and distinct head-related transfer functions among children up to about 7 years of age, mean that young children may experience a different spectral distribution of environmental sound compared to adults [29]. Applying an adult-standard weighting system to young children may therefore obscure aspects of exposure that are more relevant to their auditory experience. Future studies should aim to develop child adapted weighting methods to better capture children's true auditory exposure.

Comparison between personal and stationary dosimetry in this study also demonstrated important differences. Personal dosimeters consistently recorded higher $L_{AeqTi}$ and $L_{AFmax,95}$ values than stationary devices across all percentiles. For example, mean $L_{AeqTi}$ values were 80 dB for personal dosimeters and 70 dB for stationery, while mean $L_{AFmax}$ values were 97 dB and 81 dB, respectively. These substantial differences likely reflect variations in measurement placement and responsiveness to nearby sound sources. Personal dosimeters, worn on the shoulder near the ear, are more sensitive to moment-to-moment fluctuations in noise, such as speech or play in close proximity. Stationary microphones, on the other hand, tend to capture room-averaged background levels. This underscores the importance of selecting measurement approaches that best reflect the actual exposure experienced by children in preschool settings.

These findings support the argument made by Loh et al. [7] that fixed-location measurements may not reflect the acoustic experiences of children. While both approaches are useful, the observed differences highlight the importance of measurement choice when assessing children's sound environments. Binaural recordings or head-and-torso simulators (HATS) provide perceptually realistic data by capturing anatomical filtering effects of the outer ear and head, although their application for full-day monitoring in dynamic settings remains challengingPersonal dosimetry offers a practical, field-compatible method for capturing individual exposure patterns throughout the day. Combining these complementary methods in future studies could enhance characterisation of the preschool sound environment experienced by children.

Taken together, these results highlight the complexity of studying environmental noise effects on auditory function in preschool children. While time-of-day and weekday variation in pDPOAE amplitudes was observed, individual exposure levels did not predict response magnitude, suggesting that transient physiological factors, methodological constraints, or subtle effects not captured by current metrics may be more influential. The primary aim was to examine group-level patterns, and the use of linear mixed models allowed us to account for repeated measures and within-subject variability. This statistical approach improves the reliability of group-level estimates; the study was not designed or powered to examine individual-level changes or susceptibility. Future research should consider designs that allow for the detection of individual risk factors or atypical responses to noise exposure in young children.

## Future directions

While this study provides valuable insights into the relationship between preschool noise exposure and cochlear function, further research is required to clarify the dose-response relationship between noise exposure and cochlear function in young children. Future studies should prioritise the development of child-specific reference data for the stability of pDPOAE amplitudes, particularly under pressurised conditions. Given the complexity of preschool environments, improved methods for noise assessment are needed. Personal dosimetry should remain the preferred approach for accurately capturing children's real-world noise exposure, due to its sensitivity to moment-to-moment fluctuations. In addition, frequency-specific analyses to capture noise exposure characteristics more relevant to children's auditory experiences, considering that A-weighted sound levels may not reflect their full auditory experience due to anatomical differences.

Ethical constraints preclude controlled experimental designs in preschool-aged children; therefore, longitudinal observational studies are warranted to assess the cumulative effects of noise exposure in both preschool and home settings. Such designs would better reflect real-world exposure patterns and associated risks.

In addition, middle-ear function plays a significant role in the reliability of OAE measurements. Further work is needed to develop improved methods for middle-ear monitoring and dynamic pressurisation, particularly in young children with unstable or fluctuating middle-ear pressures. Longitudinal research will also be crucial to understand the cumulative effects of noise exposure on cochlear function over time.

Finally, future studies should incorporate strategies to detect both population-level patterns and individual-level variability in cochlear responses. Group analyses provide robust population estimates but may obscure meaningful changes in more susceptible children. Analytical approaches capable of identifying outliers or atypical responses, such as screening for extreme deviations from baseline, should complement linear mixed modelling. Larger, more diverse samples will also be required to reliably assess subtle effects and individual risk factors, including age, sex, and baseline auditory status.

## Conclusion

This study underscores the challenges and complexities of studying noise exposure effects on cochlear function in preschool-aged children. Despite observing significant variation in pDPOAE amplitudes across time of day and week, no direct associations were found between individual noise exposure levels and cochlear responses. This absence of significant associations may be attributable to several methodological constraints, of particular importance, including: (1) limited statistical power to detect small or transient effects; (2) the timing of measurements, as DPOAE changes are transient and

previous studies demonstrating short-term effects have typically analysed responses within 15 minutes post-exposure, whereas such precise timing relative to noise exposure was not captured here; (3) the potential influence of prior noise exposure before baseline measurements, raising questions about the adequacy of baseline status; (4) the possibility of nonlinear level-dependent effects, including thresholds below which cochlear function may not be affected; and (5) measurement variability, and the dynamic nature of cochlear function in young children.

Future research should focus on refining measurement techniques, including the use of personal dosimetry, improved frequency-specific analyses, and better methods for monitoring middle-ear function. Additionally, observational studies could offer valuable insights into the cumulative effects of real-world noise exposure in both home and preschool environments. Such approaches avoid ethical concerns and provide a practical framework for studying long-term auditory outcomes in young children.

Given the critical importance of early childhood development, longitudinal studies tracking cochlear function across time will be essential for determining whether repeated exposure to noise in preschool environments influences hearing development over the long term. By incorporating long-term follow up, improving measurement precision, and accounting for individual susceptibility, future research can help clarify the potential impacts of early noise exposure on auditory health in children.

## Supporting information

**S1 File. Supporting tables.**
(PDF)

**S2 File. Diagnostic plots.**
(PDF)

**S1 Fig. Distribution of tympanometric peak pressure (TPP) values in the left and right ear across all retained pDPOAE measurements.**
(TIF)

**S2 Fig. Change in tympanometric peak pressure (TPP) at follow-up compared to baseline, by ear and measurement round.**
(TIF)

**S3 Fig. Within-subject change in pDPOAE amplitude (dB SPL) across the preschool week.**
(TIF)

## Acknowledgments

The authors thank the Municipality of Gothenburg, as well as the preschool principals, teachers, parents, and children who participated in this study. Their engagement and cooperation in the fieldwork were invaluable. We also appreciate the support of the preschool staff in facilitating study logistics and data collection. Additionally, we are grateful to our colleagues and collaborators for their insightful discussions and feedback throughout the research process.

## Author contributions

**Conceptualization:** Loisa Sandström, Sofie Fredriksson, Elina Mäki-Torkko, Kerstin Persson Waye.

**Data curation:** Loisa Sandström.

**Formal analysis:** Loisa Sandström, Mikael Ögren.

**Funding acquisition:** Kerstin Persson Waye.

**Investigation:** Loisa Sandström.

**Methodology:** Loisa Sandström, Sofie Fredriksson, Elina Mäki-Torkko, Kerstin Persson Waye.

**Project administration:** Loisa Sandström.

**Resources:** Mikael Ögren.

**Supervision:** Sofie Fredriksson, Elina Mäki-Torkko, Kerstin Persson Waye.

**Validation:** Loisa Sandström, Mikael Ögren.

**Visualization:** Loisa Sandström.

**Writing – original draft:** Loisa Sandström.

**Writing – review & editing:** Loisa Sandström, Sofie Fredriksson, Elina Mäki-Torkko, Janina Fels, Kerstin Persson Waye.

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
