## [Decision Letter · Decision Letter 0]

2 Jul 2025

Dear Dr. Sandström,

Thank you for submitting your manuscript to PLOS ONE. After careful consideration, we feel that it has merit but does not fully meet PLOS ONE’s publication criteria as it currently stands. Therefore, we invite you to submit a revised version of the manuscript that addresses the points raised during the review process.

**ACADEMIC EDITOR:**

The reviewers have evaluated your manuscript and recommended major revisions. Should you choose to proceed with a revision, it is strongly advised that you address all the comments raised by the reviewers. In particular, please give special attention to the feedback from Reviewer 2 and Reviewer 3, as their concerns are considered the most significant. Once your revised manuscript is submitted, it will be sent back to the reviewers for further evaluation before a final decision is made.

We look forward to receiving your revised manuscript.

Kind regards,

Rohit Ravi, Ph.D.

Academic Editor

PLOS ONE

Journal Requirements:

a) If there are ethical or legal restrictions on sharing a de-identified data set, please explain them in detail (e.g., data contain potentially identifying or sensitive patient information, data are owned by a third-party organization, etc.) and who has imposed them (e.g., a Research Ethics Committee or Institutional Review Board, etc.). Please also provide contact information for a data access committee, ethics committee, or other institutional body to which data requests may be sent

3. In the online submission form you indicate that your data is not available for proprietary reasons and have provided a contact point for accessing this data. Please note that your current contact point is a co-author on this manuscript. According to our Data Policy, the contact point must not be an author on the manuscript and must be an institutional contact, ideally not an individual. Please revise your data statement to a non-author institutional point of contact, such as a data access or ethics committee, and send this to us via return email. Please also include contact information for the third party organization, and please include the full citation of where the data can be found.

4. Please amend the manuscript submission data (via Edit Submission) to include author Kerstin Persson Waye.

5. Please amend your authorship list in your manuscript file to include author Kerstin Persson Persson Waye.

Reviewers' comments:

Reviewer's Responses to Questions

**Comments to the Author**

1. Is the manuscript technically sound, and do the data support the conclusions?

Reviewer #1: Yes

Reviewer #2: Partly

Reviewer #3: Yes

2. Has the statistical analysis been performed appropriately and rigorously?

Reviewer #1: Yes

Reviewer #2: Yes

Reviewer #3: Yes

3. Have the authors made all data underlying the findings in their manuscript fully available?

Reviewer #1: Yes

Reviewer #2: Yes

Reviewer #3: Yes

4. Is the manuscript presented in an intelligible fashion and written in standard English?

Reviewer #1: Yes

Reviewer #2: Yes

Reviewer #3: Yes

Reviewer #1: The manuscript was extremely well-written and provided a good amount of detail in or for replication to occur in future studies. I have some minor comments to be addressed but do believe this is a manuscript close to ready.

Reviewer #2: This study examined the correlation between pressurized distortion product otoacoustic emissions (pDPOAEs) and sound exposure in a group of preschool children. pDPOAEs were measured in the morning and afternoon on two days (the beginning and end of a school week). Sound exposure was measured using dosimetry on two days. Results revealed no significant correlations between pDPOAE amplitude and sound exposure. There were some significant differences in pDPOAE amplitudes across time and between sexes. The authors concluded that the lack of significant correlations may be due to methodologic issues or how the cochlea recovers from sound exposure in children.

On the positive side, the work is novel and builds upon the authors' previous work (Sandström et al., 2025) by using pDPOAEs to reduce confounds of negative middle ear pressure. The methodology for sound exposure measurements are described in sufficient detail. Statistical analyses are appropriate and the authors provide detailed statistical results in figures, tables, and supplementary material. However, there were significant weaknesses that are described below, along with minor points for the authors to consider.

Major points:

1) The use of pDPOAEs can reduce the confound of negative middle ear pressure primarily at frequencies ≤2 kHz (Beck et al., 2016, Hear Rev). However, the authors only analyzed pDPOAEs from 3-8 kHz, so the advantage of using pDPOAEs over standard (unpressurized) DPOAEs would expected to be minimal. A limitation of this study is the lack of a control condition using standard DPOAEs. This would allow the authors to directly compare standard vs. pDPOAEs to determine if there is an advantage of pDPOAEs for the current study.

2) DPOAE amplitudes decrease as middle ear pressure becomes more negative (e.g., Thompson et al., 2015, Ear Hear). The authors excluded participants with tympanometric peak pressures falling outside of +50 to -150 daPa, but they did not report the tympanometric peak pressures of the included participants. Therefore, the extent to which pDPOAEs were advantageous in this study cannot be determined. If most participants had tympanometric peak pressures near 0 daPa, the results of pDPOAEs should be similar to standard (unpressurized) DPOAEs.

3) The authors found some differences in pDPOAE amplitudes across time. However, there were no measurements of immediate test-retest reliability of pDPOAEs. Therefore, it is unclear if these differences in amplitude fell within measurement variability or represent true changes in cochlear function.

Minor points:

1) The Introduction lacks sufficient background on DPOAEs (e.g., how they are generated and measured) and the differences between pDPOAEs and DPOAEs.

2) Throughout the manuscript, the authors refer to the pDPOAEs as measures of "hearing function." This is not accurate because pDPOAEs are a physiologic measure of outer hair cell function and not a measure of auditory perception. Also, pDPOAE frequencies should be referred to as "f2 frequency" rather than "hearing frequency" (e.g., the x-axis label of Fig. 3).

3) L434: It appears that "processed" should be replaced with "pressurized." Consider clarifying.

4) L435: It is unclear how pDPOAE amplitudes can be expressed in units of dB(A). Consider clarifying.

5) L437-438: Note that the left panel of Fig. 4 shows the right ear results and the right panel shows the left ear results.

6) L530: Note that Tables S3-S5 are referenced in the text before Table S2.

Reviewer #3: General Comments

The authors conducted a study that has important implications for early detection of sound induced auditory dysfunction in young children. The methodology includes several strengths including dosimetry documentation of the duration and dose of noise exposure, the use of a highly sensitive measure cochlear (outer hair cell) function (DPOAEs), and a DPOAE measurement technique that accounts for middle ear pressure as defined by tympanometry. There are several serious weaknesses in the study methodology including limited baseline information about the auditory status of participants and the brief (one week) period of data collection. The authors might consider a similar study that eliminates these weaknesses, and that focuses on an older (adolescent, teenage, and/or college) student population.

Specific Comments

- More detailed information about the study participants should be provided. For example, did any of the children have perinatal risk factors for auditory, and specifically cochlear, dysfunction (e.g., admission to a neonatal intensive care unit, perinatal infection, ototoxic drugs, family history)?

- Was pure tone and speech audiometry performed and, if so, what were the findings? If not, why wasn’t hearing status fully evaluated before or at the beginning of the study?

- Was there any attempt to document the participants’ exposure to noise or music outside of school hours, e.g., the child’s use of personal sound devices?

- The authors mention in the Discussion section the possibility of “selection bias.” The rationale should be explained for exclusion criteria that involved DPOAE findings, e.g., “Measurements were excluded if the DPOAE amplitude was below −10 dB SPL, if the signal-to-noise ratio was less than 6 dB, or if measurement reliability was below 80%”. It would be useful to know how many potential participants were excluded because of these criteria. Is it possible that children who are susceptible to noise or music induced cochlear dysfunction were excluded from the study because of these criteria or, as the authors state, "skewing the ... samples toward children with more stable physiology"?

- Visual analysis of possible DPOAE changes from the beginning to the end of the week (Figure 3) would be simplified if right versus left ear data were plotted on separate graphs, and composite DPOAE amplitudes recorded at the beginning versus end of the week were represented with different colors and plotted on the same graph. The authors may also consider another similar figure but with DPOAE amplitudes for the beginning versus the end of the week displayed as a scatter plot.

- Exclusive analysis of group data may obscure possible noise effects for individual participants. Did any of the participants show meaningful changes in DPOAE amplitudes over time?

- The authors state in the Discussion section that “This study investigated how real-world preschool noise exposure relates to cochlear function in young children”. However, potential effects of classroom noise on cochlear function are likely to be chronic, occurring over months and years rather than days. Why were data collected only over a one-week time frame, rather than over the course of a semester or a school year? The authors raise this possibility in the Discussion section: “It is also possible that effects occurred prior to the measurement session or require longer follow-up periods cochlear function in young children to be captured.” The authors should consider further discussion of this important point.

**Do you want your identity to be public for this peer review?** For information about this choice, including consent withdrawal, please see our Privacy Policy

Reviewer #1: No

Reviewer #2: No

Reviewer #3: No

---

## [Author Response · Author response to Decision Letter 1]

15 Aug 2025

Response to Reviewer Comments

Manuscript Title: Impact of Preschool Sound Exposure on Hearing Function in Young Children: An Analysis of Pressurised Distortion Product Otoacoustic Emissions

UPDATED, Manuscript Title: Short-Term Impact of Preschool Sound Exposure on Outer Hair Cell Function in Young Children: An Analysis Using Pressurised Distortion Product Otoacoustic Emissions

Manuscript Number: PONE-D-25-17569R1

Journal: PLOS One

Date: 20250708

Corresponding Author: Loisa Sandström, Loisa.Sandstrom@gu.se

General Statement

We thank the editor and reviewers for their thoughtful and constructive feedback. We have carefully addressed each point and revised the manuscript accordingly. In addition to responding to all scientific comments, we have made the necessary updates to ensure full compliance with PLOS ONE’s editorial and data policies, including manuscript formatting, author listing, data availability, and Supporting Information captions.

Below, we provide a point-by-point response.

• Section 1 includes responses to requests from the Editorial Office.

• Section 2 contains our detailed replies to each reviewer comment.

In each case, reviewer/editor comments are shown in bold (and as standalone headings in Section 2), followed by our response and a description of any changes made to the manuscript.

Please note that all line numbers referenced in this rebuttal refer to the version of the manuscript with tracked changes not visible.

Editorial Office Requests

We thank the editorial team for these clear and detailed instructions. Below we outline the steps taken to ensure compliance with all editorial and data requirements:

1. Manuscript Formatting

The manuscript file has been reviewed and revised to follow PLOS ONE's style requirements, including formatting and file naming conventions, as per the provided templates: Manuscript Body Formatting Guidelines and Title, Author, Affiliations Formatting Guidelines

2. Data Availability and Ethics

Response:

Thank you for highlighting this important clarification. We have now revised the Data Availability Statement in the manuscript and submission system to comply with PLOS ONE’s data policy. Due to ethical restrictions outlined in our approved protocols (Swedish Ethical Review Authority Dnr: 2019-03412, 2020-03944, 2022-03580-02), we are not permitted to share raw individual-level data on children's hearing and noise exposure publicly.

A minimal anonymised dataset, excluding directly identifiable or sensitive information, will be therefore made available to qualified researchers upon request. The Swedish National Data Service (SND) now serves as the institutional, non-author contact point for these requests. Full contact details and a persistent link to the dataset description will be included upon final archiving. The submission form and manuscript have been updated accordingly.

Updated Data Availability Statement (for manuscript and submission system)

The data underlying the results presented in this study cannot be shared publicly owing to ethical restrictions on research involving children, as specified in approvals from the Swedish Ethical Review Authority (Dnr 2019-03412, 2020-03944, 2022-03580-02). A minimal anonymised dataset (excluding directly identifiable or sensitive information) has been deposited with the Swedish National Data Service (SND) and is available to qualified researchers who meet the criteria for access to confidential data. The dataset is available at https://doi.org/10.5878/kp2z-2239. For further enquiries, or to request access to the data, email request@snd.se.

3. Author Addition

Co-author Kerstin Persson Waye has been added in the online submission system with appropriate authorship contributions. We have carefully reviewed both the manuscript and the submission metadata to ensure all co-authors are listed correctly. If there is still an oversight on our part, we would be grateful for clarification and will promptly make any necessary corrections.

4. Supporting Information Captions

Captions for all Supporting Information files (S1–S15 Figures and S1–S7 Tables) have now been added at the end of the manuscript. All in-text citations have been reviewed and updated to ensure consistency with the corresponding Supporting Information file labels and order.

We hope that these updates address all outstanding requirements and are happy to provide further clarification if needed.

Reviewers Comments

Reviewer 1:

1.1 Line 30: DPOAE should be in parentheses here. ¨

Response:

Corrected in the revised manuscript.

1.2 Line 37: pDPOAE or DPOAE?

Response:

This has been corrected as it should be pDPOAE, thank you for pointing this out.

1.3 Line 265 and 298: Here (and on 298), military time is used but on line 171, it is not. Pick one and stay consistent through the manuscript.

Response:

Thank you. Time formatting has been standardised to “military time”, 24h clock, throughout the manuscript.

1.4 Line 434: I thought the p in pDPOAE stood for pressurised, not processed.

Response:

The wording has been corrected to pressurised.

1.5 Table 4: Remove the ‘i’ in measurements.

Response:

This has been corrected.

The introduction is well-written and lays out the purpose for conducting this study well.

Response: Thank you!

1.6 Line 203 Hearing Assessment Section: pDOAEs are not well known. Please explain in more detail how it compensates for middle ear pressure variations. Additionally, please indicate for the readers unaware of DPOAEs what higher amplitudes suggest compared to low (e.g., good OHC function above the noise floor).

Response:

Thank you for this helpful comment. We have expanded the background with regards to the Methods section (Lines 231–235) to further explain pressurised DPOAEs (pDPOAEs).

To address the second part of your comment, we added an explanation in the Discussion (Lines 535–537). We have also added a new section introducing DPOAEs more clearly in the Introduction (Lines 119–129) and expanded the Introduction (Lines 143–149) to provide further context on pDPOAE and its methodological advantages.

Line 277-278: data excluded for lunchtime and outdoor periods. Is there also a “quiet time” or “nap time” that was also accounted for?

Response:

Thank you for this important observation. Nap time was not a factor in this dataset, as the participating departments catered to older preschool children who did not have scheduled naps. Instead, the early afternoon typically involved calmer activities. These periods were not excluded, as they reflect the natural variation in preschool routines. To clarify this point, we have added information the Methods section (Lines 308–310).

1.7 Throughout the manuscript, the acronym changes between pDPOAE and DPOAE. Please review all instances and make sure they are reporting the correct one.

Thank you. We have reviewed the manuscript and ensured consistent, correct use of DPOAE and pDPOAE throughou

1.8 Line 350: Which citation are you referencing for these interpretation values of ICC?

Response:

Thank you for pointing this out, the correct reference has been added to the manuscript (on line 382):

Koo, T. K., & Li, M. Y. (2016). A Guideline of Selecting and Reporting Intraclass Correlation Coefficients for Reliability Research. Journal of chiropractic medicine, 15(2), 155–163. https://doi.org/10.1016/j.jcm.2016.02.012¨

1.9 Figure 4: Include r and p values in the figure (usually top left) for each ear

Response:

We thank the reviewer for this helpful suggestion. Figure 4 has been updated accordingly.

1.10 Line 427 & 440: I am confused here. It is reported that there are no associations but then later is says there are potential associations. Please clarify.

Response:

Thank you for this helpful observation. We agree that the original wording in the figure legend could cause confusion between visual trends in the scatter plots and the main inferential results. In the previous version, “potential associations” referred to the visual pattern of points and the descriptive Pearson r values shown in Figure 4, not to statistically significant effects. As stated in the Results, inferential statistics from the linear mixed models did not identify significant associations between individual sound exposure and pDPOAE amplitudes. We have therefore revised the figure legend (Lines 480–490) to make this distinction explicit, clarifying that the Pearson correlations are presented for descriptive purposes only and that the main results are based on linear mixed models.

1.11 Line 457/462/468: significantly higher means better here, correct? Please include interpretation.

Response:

Thank you for this important comment. Yes, higher DPOAE amplitudes are generally interpreted as reflecting better outer hair cell responsiveness. In line with reporting conventions, we have kept the Results section strictly descriptive and have therefore not included this interpretation there. In order to address your point, we now explicitly state in the Discussion (Lines 533–537) that higher amplitudes typically indicate better OHC responsiveness, whereas reductions are generally interpreted as consistent with temporary changes in OHC function. This interpretation is also supported by the revised Introduction (Lines 124–129), which notes that DPOAE amplitudes can decrease following noise exposure, indicating a temporary reduction in OHC function that may suggest early signs of cochlear stress.

1.12 Line 538: add the referenced study citation here.

Response:

Thank you for noting this omission. Please note that the section where the reference originally appeared has been revised following other reviewer comments and no longer appears in its previous form, the citation appears on the lines: 595, 612, 638

The correct citation is:

Thompson S, Henin S, Long GR. Negative middle ear pressure and composite and component distortion product otoacoustic emissions. Ear Hear. 2015;36: 695–704. doi:10.1097/AUD.0000000000000185

1.4 Line 570: Please explain why A-weighting was used given these limitations. Either here or in the Methods section.

Response:

Thank you for this important comment. We agree that clarification is needed. A-weighting was used because it remains the internationally recognised standard for environmental and occupational noise measurement and is widely used in studies of preschool environments. While we acknowledge that A-weighting is based on adult hearing sensitivity and may not fully capture how young children perceive sound it enables comparability with previous research and supports alignment with regulatory thresholds and guidelines. This rationale has now been added to the Discussion section on lines 685-687.

1.13 Future Directions: It would also be interesting to see if there is a relationship between personal dosimetry/DPOAE and teacher performance reports. That is, are those with higher dosimetry results also those that have more behavioral or attention problems seen by the teacher?

Response:

Thank you for this thoughtful suggestion. While formal teacher evaluations are not part of the pedagogical mission in Swedish preschools, we did collect behavioural data using the Strengths and Difficulties Questionnaire (SDQ), completed by preschool staff. These data are part of a broader longitudinal study and will be analysed in relation to other environmental factors, including room acoustics and longer-term exposure patterns.

In the present study, our focus was limited to the short-term effects of noise exposure on cochlear function, using pDPOAE as a physiological outcome measure. Including behavioural outcomes such as those from the SDQ would require a different analytical framework and longer follow-up. We agree that a broader perspective is important and consider the SDQ data a valuable resource for future work within the longitudinal project.

Reviewer 2

Major points:

2.1 The use of pDPOAEs can reduce the confound of negative middle ear pressure primarily at frequencies ≤2 kHz (Beck et al., 2016, Hear Rev). However, the authors only analyzed pDPOAEs from 3-8 kHz, so the advantage of using pDPOAEs over standard (unpressurized) DPOAEs would expected to be minimal. A limitation of this study is the lack of a control condition using standard DPOAEs. This would allow the authors to directly compare standard vs. pDPOAEs to determine if there is an advantage of pDPOAEs for the current study.

Response:

We appreciate this thoughtful comment and agree that the benefits of pressurised DPOAEs (pDPOAEs) are most pronounced at lower frequencies (≤ 2 kHz). However, our focus on the 3–8 kHz range was intentional, as these higher frequencies are more sensitive to early noise-induced cochlear changes, which aligns with the study’s aim to investigate potential short-term effects of preschool noise exposure on outer hair cell function. Although Sun and Shaver (2009) reported that the amplitude-enhancing effect of positive middle-ear pressure was greatest at ≤ 2 kHz, they also observed measurable improvements at 3 kHz, indicating that middle-ear status can still influence responses in this range.

A direct comparison between standard and pressurised DPOAEs was beyond the intended scope of this study. Our objective was to implement a robust, field-feasible protocol optimised for young children. Given the high prevalence of mild negative middle-ear pressure in this population, pressurisation was applied across all tested frequencies to minimise variability between sessions, even at higher frequencies where the benefit is smaller. Importantly, previous studies have not reported adverse effects of pressurisation at higher frequencies, supporting its uniform application in the present protocol. We have clarified this rationale and justification in the revised Discussion (lines 599–609).

2.2 DPOAE amplitudes decrease as middle ear pressure becomes more negative (e.g., Thompson et al., 2015, Ear Hear). The authors excluded participants with tympanometric peak pressures falling outside of +50 to -150 daPa, but they did not report the tympanometric peak pressures of the included participants. Therefore, the extent to which pDPOAEs were advantageous in this study cannot be determined. If most participants had tympanometric peak pressures near 0 daPa, the results of pDPOAEs should be similar to standard (unpressurized) DPOAEs.

Response:

We thank the reviewer for raising this point. The revised Results (lines 446–452) now report detailed tympanometric peak pressure (TPP) data for all included recordings. Nearly all TPP values were between –100 and +50 daPa, with about half in the –49 to –1 daPa range (see S3 Table; Supplementary Fig. S1). Thus, a substantial proportion of sessions exhibited mild-to-moderate negative middle-ear pressure, where unpressurised DPOAEs would likely have been attenuated. For sessions with TPP near 0 daPa, pDPOAE and standard DPOAE would yield comparable results.

We also included paired baseline and follow-up TPP values for each child and ear in Fig. S2, showing that many individual measurements differed between the two timepoints on the same day. These patterns indicate that negative middle-ear pressure was common in this cohort, further supporting the decision to use pDPOAEs to reduce variability. These new analyses and rationale are now reflected in the Methods (lines 392–397), Results (lines 446–452), and Discussion (lines 598–602).

2.3 The authors found some differences in pDPOAE amplitudes across time. However, there were no measurements of immediate test-retest reliability of pDPOAEs. Therefore, it is unclear if these differences in amplitude fell within measurement variability or represent true changes in cochlear function.

Response:

We thank the reviewer for highlighting this issue. Immediate test–retest was not undertaken. Pressurised DPOAE in 4–6-year-olds requires tympanometry, pressure matching and stable probe placement. Adding immediate repeats would extend sessions and increase participant burden in a field protocol that already included up to four measurements per child in one week. This falls outside the study’s objective, which was to examine short-term changes in relation to real-world exposure. We now state this explicit

---

## [Decision Letter · Decision Letter 1]

3 Sep 2025

Dear Dr. Sandström,

Thank you for submitting your manuscript to PLOS ONE. After careful consideration, we feel that it has merit but does not fully meet PLOS ONE’s publication criteria as it currently stands. Therefore, we invite you to submit a revised version of the manuscript that addresses the points raised during the review process.

**ACADEMIC EDITOR:**

I have received the comments from both reviewers, and I am pleased to inform you that the feedback is positive. However, as mentioned in your previous communication, a few minor changes are still required in the version you uploaded. I kindly request you to make the necessary edits and resubmit the final version at your earliest convenience so that we may proceed further.

Thank you for your cooperation.

We look forward to receiving your revised manuscript.

Kind regards,

Rohit Ravi, Ph.D.

Academic Editor

PLOS ONE

Journal Requirements:

Reviewers' comments:

Reviewer's Responses to Questions

**Comments to the Author**

Reviewer #1: All comments have been addressed

Reviewer #3: All comments have been addressed

2. Is the manuscript technically sound, and do the data support the conclusions?

Reviewer #1: Yes

Reviewer #3: Yes

3. Has the statistical analysis been performed appropriately and rigorously?

Reviewer #1: Yes

Reviewer #3: Yes

4. Have the authors made all data underlying the findings in their manuscript fully available?

Reviewer #1: Yes

Reviewer #3: Yes

5. Is the manuscript presented in an intelligible fashion and written in standard English?

Reviewer #1: Yes

Reviewer #3: Yes

Reviewer #1: All of my comments were addressed adequately. I have no further concerns regarding this manuscript submission. I believe it is well-written and explains the process well in addition to the findings.

Reviewer #3: The authors have diligently responded to this reviewer's comments and concerns. The revisions are adequate.

**Do you want your identity to be public for this peer review?** For information about this choice, including consent withdrawal, please see our Privacy Policy

Reviewer #1: No

Reviewer #3: No

---

## [Author Response · Author response to Decision Letter 2]

3 Sep 2025

Dear Editor,

Thank you again for the positive feedback and the opportunity to revise our manuscript. In addition to the changes previously requested, I have made a few additional minor edits to improve clarity and correctness. These include:

Correction of small word-level errors identified during a final review.

Removal of a paragraph that was mistakenly retained in the previous version (lines 723–728), which should have been removed.

I have uploaded both the revised manuscript and the version with tracked changes, and the files are named with today’s date for clarity.

These adjustments do not affect the scientific content or conclusions of the manuscript but enhance its readability and presentation.

Please let me know if any further clarification is needed.

Kind regards,

Loisa Sandström

---

## [Editor Report · Decision Letter 2]

5 Sep 2025

Short-Term Impact of Preschool Sound Exposure on Outer Hair Cell Function in Young Children: An analysis Using Pressurised Distortion Product Otoacoustic Emissions

PONE-D-25-17569R2

Dear Dr. Sandström,

We’re pleased to inform you that your manuscript has been judged scientifically suitable for publication and will be formally accepted for publication once it meets all outstanding technical requirements.

Kind regards,

Rohit Ravi, Ph.D.

Academic Editor

PLOS ONE

---

## [Editor Report · Acceptance letter]

PONE-D-25-17569R2

PLOS ONE

Dear Dr. Sandström,

I'm pleased to inform you that your manuscript has been deemed suitable for publication in PLOS ONE. Congratulations! Your manuscript is now being handed over to our production team.

Kind regards,

on behalf of

Dr. Rohit Ravi

Academic Editor

PLOS ONE